# Platelet Abnormalities in Children with Laboratory-Confirmed Influenza

**DOI:** 10.3390/diagnostics13040634

**Published:** 2023-02-08

**Authors:** August Wrotek, Oliwia Wrotek, Teresa Jackowska

**Affiliations:** 1Department of Pediatrics, Centre of Postgraduate Medical Education, Marymoncka 99/103, 01-813 Warsaw, Poland; 2Department of Pediatrics, Bielanski Hospital, Cegłowska 80, 01-809 Warsaw, Poland; 3Student Research Group, Bielanski Hospital, Cegłowska 80, 01-809 Warsaw, Poland

**Keywords:** platelets, influenza, virus, mean platelet volume, thrombocytopaenia, thrombocytosis, platelet indices, pneumonia, otitis media

## Abstract

Background: The role of platelets in the immune response against influenza has been raised, and a diagnostic or prognostic value of platelet parameter abnormalities, including platelet count (PLT), or mean platelet volume (MPV), has been suggested. The study aimed to analyze the prognostic value of platelet parameters in children hospitalized due to laboratory-confirmed influenza. Methods: We retrospectively verified the platelet parameters (PLT, MPV, MPV/PLT, and PLT/lymphocyte ratio regarding the influenza complications (acute otitis media, pneumonia, and lower respiratory tract infection—LRTI), and the clinical course (antibiotic treatment, tertiary care transfer, and death). Results: An abnormal PLT was observed in 84 out of 489 laboratory-confirmed cases (17.2%, 44 thrombocytopaenia cases, and 40 thrombocytoses). Patients’ age correlated negatively with PLT (rho = −0.46) and positively with MPV/PLT (rho = 0.44), while MPV was not age-dependent. The abnormal PLT correlated with increased odds of complications (OR = 1.67), including LRTI (OR = 1.89). Thrombocytosis was related to increased odds of LRTI (OR = 3.64), and radiologically/ultrasound-confirmed pneumonia (OR = 2.15), mostly in children aged under 1 year (OR = 4.22 and OR = 3.79, respectively). Thrombocytopaenia was related to antibiotic use (OR = 2.41) and longer hospital stays (OR = 3.03). A lowered MPV predicted a tertiary care transfer (AUC = 0.77), while MPV/PLT was the most versatile parameter in predicting LRTI (AUC = 0.7 in <1 yo), pneumonia (AUC = 0.68 in <1 yo), and antibiotic treatment (AUC = 0.66 in 1–2 yo and AUC = 0.6 in 2–5 yo). Conclusions: Platelet parameters, including PLT count abnormalities and MPV/PLT ratio, are related to the increased odds of complications and a more severe disease course, and may add important data in assessing pediatric influenza patients, but should be interpreted cautiously due to age-related specificities.

## 1. Introduction

Influenza is one of the most important viral pathogens that may cause a varied disease course; from a self-limiting illness to severe complicated cases, requiring intensive care, or resulting in a fatal outcome [1]. The influenza attack rate in children may exceed 15% [2], and the most severe disease course is related to a younger age; rough estimates report over 10 million cases of acute lower respiratory tract infection (ALRI) in children under 5 years of age, and around 870,000 influenza-associated ALRI hospitalizations [3]. The risk of complications related to influenza is high, with acute otitis media (AOM) and pneumonia, being the most frequent sequelae, found in 22–30% and 10–12% of hospitalized cases, respectively [4]. Although there is a possibility of a causative antiviral treatment (neuraminidase inhibitors, NAIs), which decreases the number of complications, including AOM [5,6,7,8,9] or pneumonia [7,10], and respiratory complications other than pneumonia [8,9], yet, early implementation of the treatment (optimally within 48 h since the first symptoms) is crucial for shortening the duration of symptoms [5,6,7,10]. One of the major obstacles in influenza treatment is the risk of bacterial suprainfections. The reported frequency of bacterial coinfection varies hugely, from 2% to 65% [11]. *Streptococcus pneumoniae* and *Staphylococcus aureus* are the most frequent bacteria responsible for suprainfections (accounting for 35% and 28% of cases, respectively), while the remaining cases are caused by a wide spectrum of bacterial pathogens [11]. Therefore, a high antibiotic prescription rate is observed; during the 2009 pandemic, antibiotics were given in approximately 45% of hospitalized pediatric cases [12], although in some settings the rate exceeded 75% [13].

The immune response against influenza infection is of particular interest, and the role of platelet activation in the host defense is being investigated. The platelets seem to internalize the influenza virus, and a cross-communication between the platelets and the neutrophils mediates the immune response, including a complement path [14]. However, only a limited number of studies focused on platelet parameters in the course of influenza, and the analyzes mostly included the platelet count (PLT), the mean platelet volume (MPV), the mean platelet volume/platelet count ratio (MPV/PLT), the lymphocyte*platelet ratio (LYM*PLT), or the platelet to lymphocyte ratio (PLT/LYM) [15,16,17,18,19,20]. 

The platelet count acts differently under specific conditions, depending, inter alia, on the aetiological factor [19]. In influenza and several other viral infections, a general tendency towards a decreased PLT has been observed, although some exceptions may be pointed out (the respiratory syncytial virus, RSV, for example) [19]. The role of platelet activation in bacterial infections, especially pneumococcal pneumonia, has been shown more precisely. Streptococcus pneumoniae promotes a systemic activation of platelets by various pathways, including pneumolysin, hydrogen peroxide, and bacterial adhesins [21]. Alterations in the platelet count presented a prognostic value, with special emphasis on thrombocytopaenia [21]. In terms of influenza complications, a study on children hospitalized due to A/H1N1 pandemic influenza revealed higher platelet counts in patients who developed pneumonia [16]. An increased platelet-to-lymphocyte ratio (PLT/LYM) has been suggested to have a diagnostic value in distinguishing community-acquired pneumonia (CAP) from healthy controls [17]. Moreover, it showed a potential prognostic value, since inpatients had a higher PLT/LYM compared to outpatients [17]. The platelet count may be useful in differentiating between influenza and a bacterial coinfection. In a series of patients hospitalized due to AH1N1 and CAP, higher PLT was observed in those with bacterial coinfections [15].

Another key parameter is the mean platelet volume, alongside with the mean platelet volume/platelet count ratio (MPV/PLT), as the MPV correlates with platelet activation [20]. High MPV/PLT (together with low lymphocyte*platelet; LYM*PLT, low) has been reported to have a diagnostic value for distinguishing children with influenza-like illness symptoms caused by influenza type A, as compared to non-influenza cases [18]. With regards to pneumonia, a decreased MPV was observed in ill children in comparison to healthy controls, but it was significantly higher among inpatients than outpatients with pneumonia [20].

Research to date is scarce and contains a rather low number of pediatric patients, and there is a definite need for improvements in influenza diagnostics. In this study, we sought to analyze the prognostic value of platelet parameters (PLT, MPV, MPV/PLT, and PLT/LYM) in a huge group of children hospitalized due to influenza with regards to the prediction of complications (AOM, pneumonia, and lower respiratory tract infection—LRTI), and the clinical course of influenza (the need for antibiotic treatment, the need for tertiary care transfer, and death).

## 2. Materials and Methods

The study was granted permission by the local Ethics Committee at the Centre of Postgraduate Medical Education, Warsaw (permission number 91/PB/2020 issued on 15 June 2020). The study was conducted in accordance with the Declaration of Helsinki with its later amendments, and parents’/tutors’/patient’s consent was waived due to its retrospective character. 

The electronic medical charts of children hospitalized at the Department of Pediatrics (aged 0–18 years old), Bielanski Hospital, Warsaw, between January 2013 and March 2020 were verified. The search aimed to retrieve patients with one of the ICD-10 (International Classification of Diseases, 10th Revision) codes corresponding to influenza: J10 or J11 with extensions. Only children with laboratory-confirmed influenza were eligible for the study; the diagnosis was confirmed with the rapid influenza diagnostic test (RIDT), and/or reverse transcription-polymerase chain reaction (RT-PCR), performed in samples from a nasopharyngeal swab. The RT-PCT was a reference method and was considered conclusive in dubious cases, since both false positive and false negative results of RIDT may occur [22,23,24]. All the patients diagnosed with influenza were treated with oral oseltamivir according to official Centers for Disease Control and Prevention (CDC) guidelines, i.e., children younger than 1 y old received 3 mg/kg/dose twice daily, while those aged 1 year or more received a dose appropriate for the weight (30 mg if ≤15 kg, 45 mg if >15 to 23 kg, 60 mg if >23 to 40 kg, 75 mg if >40 kg) twice a day; the standard duration of the treatment is 5 days [25]. 

Patients with a complete blood count were included in the analysis. The blood count analysis was performed with the use of Sysmex XN1000/Sysmex XN550 (Sysmex Corporation, Kobe, Japan) since 10 April 2014, and before that date, with the use of Sysmex XT2000i (Sysmex Corporation, Kobe, Japan). CRP was measured with the Cobas 6000 analyzer (Roche Diagnostics Ltd., Rotkreuz, Switzerland) with a limit of detection of 0.1 mg/L, while procalcitonin was measured with the Cobas e411 analyzer (Roche Diagnostics Ltd., Rotkreuz, Switzerland) until 4 August 2016, and with the Cobas 6000 analyzer (Roche Diagnostics Ltd., Rotkreuz, Switzerland) from 5 August 2016; the limit of detection in both cases was 0.02 ng/mL. Capillary blood gas analysis was performed with the use of the Roche Cobas b121 and b221 analyzer (Roche Diagnostics Ltd., Switzerland) until 26 January 2016, and with the RAPIDLab 348EX Blood Gas System (Siemens Healthcare Diagnostics, Erlangen, Germany) from 27 January 2016. All the analytic processes were conducted according to the manufacturers’ instructions.

The exclusion criteria consisted of: an immune deficiency (congenital or acquired), immunosuppression (e.g., children after chemotherapy or receiving an ongoing immunosuppressive therapy), a history of neoplasm, a hemodynamically significant heart disease, and a discharge on patient’s/parent’s/tutor’s demand. For the purposes of analyzing the presence of pneumonia, the lack of a radiographic confirmation of pneumonia was considered an exclusion criterion. 

Baseline characteristic assessments included the age, gender, duration of symptoms prior to hospitalization, and, separately, duration of fever prior to hospitalization. The following platelet parameters were analyzed: platelet count (PLT), mean platelet volume (MPV), MPV to PLT ratio, and PLT to lymphocyte (PLT/LYM) ratio. In line with the previously published studies in this field, thrombocytosis was defined as a platelet count > 450,000/mm^3^, while thrombocytopaenia was <150,000/mm^3^ [19].

Both age and influenza severity might be expected to influence platelet parameters, thus the analyzes were performed in the whole study group, and patients were furtherly divided into age subgroups. Firstly, due to practical implications, influenza risk groups were taken into consideration: children under 5 years old, and especially those younger than 2 years old are at higher risk of a more severe disease course, and corresponding age groups were created [25]. Moreover, based upon the previously published data on platelet parameters in children and an emphasis put on children under 1 year of age (inter alia, the highest frequency of platelet count abnormalities seen in those under 12 months of age), finally, the following age groups were created: children under 12 months of age, 12–23 months of age, 24–59 months old, and 60 or more months [19,26,27]. Nevertheless, it needs to be underlined that age-related changes in platelet parameters are rather continuous than abrupt, and the age division does not strictly reflect physiological changes [28]. 

Patients were analyzed in terms of the presence of complications: AOM, LRTI, and radiologically-confirmed pneumonia. All the endpoints, i.e., AOM, LRTI, and pneumonia were defined according to the official Polish guidelines [29]:

Acute otitis media was defined as the presence of acute signs and/or symptoms of acute otitis media (fever of at least 38 degrees Celsius, pain) and an otoscopy result: the presence of fluid in the tympanic cavity (seen as fluid level in otoscopy, a bulge of the tympanic membrane, decreased tympanic membrane mobility in pneumatic otoscopy, fluid discharge) and a reddening of the tympanic membrane. 

LRTI was defined as a final diagnosis of bronchitis, bronchiolitis, or pneumonia. Bronchitis is recognized when cough (productive or unproductive) is accompanied by wheezing or rales on auscultation, while bronchiolitis is defined as the first episode of the restriction of bronchioles (wheezing, rales on auscultation, expiratory dyspnoea, and/or hypoxia). Pneumonia was defined as the presence of clinical signs and/or symptoms (cough, fever of at least 38 degrees Celsius, tachypnea, intercoastal spaces retractions, a dull percussion, or a presence of bronchial murmur or crackles on auscultation). Tachypnea was defined as >60 breaths/minute in neonates (0–1 months), >50 breaths/minute in children 1–12 months, >40 breaths/minute in those aged 12–59 months, and >25 breaths/minute in children aged 60 months or older. 

For the purposes of the pneumonia analysis, we included only patients with a confirmation in a chest X-ray or lung ultrasound, meaning the presence of any of the following: consolidation, linear densities, patchy densities, parenchymal infiltrates, or pleural effusion or (in the case of the lung ultrasound), the presence of hypoechogenic lesions, pleural line abnormalities, bronchogram sign, or an impaired lung respiratory mobility (absence or decrease in “lung sliding”).

A second group of endpoints consisted of the clinical course assessment, i.e., length of stay (LOS), need for antibiotic treatment, need for intensive care unit (ICU), or other tertiary care reference center transfer related to the disease severity, and an unfavorable outcome. The LOS values were dichotomized upon the median value in order to be presented in a binary mode.

The analysis of the confounding factors included gender and hydration status. Hydration status was assessed by pH, bicarbonate, and base excess levels from capillary blood gas measurements, which was proved to reflect arterial blood gas value strictly [30,31]. 

The Kolmogorov-Smirnov test was used to assess data distribution, and a mean with a standard deviation (SD) or a median with an interquartile range (IQR) was used to present the results according to the data distribution. A corresponding parametric or non-parametric test (the Student *t*-test or the Mann-Whitney U test) was applied. 

The correlation between continuous data was presented as Spearmann’s rank correlation coefficient (rho). 

The receiver operating curve (ROC) analysis was used to calculate the area under the curve (AUC) with a 95% confidence interval (95%CI). Due to the age-related specificities, the ROC analysis was performed in the whole study group, and then separately in each age group in order to decrease the age influence and to calculate the most accurate cut-off values that could be used in clinical practice (they were presented only for those age groups where the statistical significance was observed). An optimal cut-off value for the prediction of the endpoints was estimated with the Youden index. In the case of statistically significant AUCs under 0.5 (meaning a destimulating, not stimulating, the effect of the verified parameter), the equation was reversed in order to present easily comparable results (i.e., AUC over 0.5), with an emphasis on the destimulating role, marked by the word “lowered” preceding the parameter. 

A multivariate regression model for dichotomous data was run to calculate the odds ratio (OR) with a 95% confidence interval. 

A *p*-value under 0.05 was considered statistically significant. The statistical analysis was conducted with the use of Statistica 13.1 software (Statsoft, Tulsa, OK, USA).

## 3. Results

During the analyzed period (87 consecutive months) there were 12,639, and 489 children were diagnosed with laboratory-confirmed influenza, corresponding to 3.9% of the total number of hospitalizations. A flowchart of the patients is shown in Figure 1. The group consisted of 262 boys and 227 girls, with a median age of 34 months (IQR: 11–63 months); the patients were divided into age groups: there were 127 infants under 12 months of age, 68 children aged 12–23 months, 157 children aged 24–59 months, and 137 children aged 5 years or more. 

There were 370 cases of influenza type A, 104 cases of influenza type B, and 15 cases of mixed influenza A and B infection, and they differed in terms of age (the lowest median in those infected with influenza type A: 30 months versus 41 months in influenza B, and 68 months in influenza A and B). A direct comparison between influenza A and B also showed a slightly shorter duration of symptoms, and a fever in those with type A (2 vs. 3 days in both cases, *p* = 0.038 and *p* = 0.033, respectively), with no differences in the platelet parameters (Table 1).

Respiratory complications were seen in 207 children (42.3%). The AOM was diagnosed in 88 cases (18%), and LRTI was diagnosed in 154 cases (31.5%), including 128 pneumonia and 26 other than pneumonia cases; 35 patients presented both types of complications, i.e., LRTI and AOM. Eighteen children were diagnosed with pneumonia on a clinical basis only, i.e., without a radiological or ultrasound confirmation, thus, they were excluded from the analysis on pneumonia. Finally, radiologically confirmed pneumonia was diagnosed in 110 out of 471 children (23.4%). 

The highest frequency of complications was observed in mixed influenza A and B infection (60%, 9 out of 15 patients), followed by type A (44%, 163 out of 370), and type B (35.6%, 37 out of 104). 

Antibiotics were implemented in 186 out of 489 children (38%), with the highest use in children coinfected with influenza A and B (46.7%, 7 out of 15), followed by type A (39.5%, 146/370), and type B (31.7%, 33/104). 

Four patients required a transfer to a tertiary care unit (one respiratory failure—ICU transfer, one pneumothorax case—the pulmonology unit, one renal failure in the course of sepsis—the nephrology unit, one infective endocarditis—the cardiology unit). The outcome was favorable in every patient. 

The median platelet count was 244 × 10^3^/µL (IQR: 190–320), and an abnormal platelet count was observed in 84 children (17.2%), including 44 cases (9% of the whole study group) of thrombocytopaenia and 40 cases (8.2%) of thrombocytosis (Table 2). Children with an abnormal platelet count did not differ in terms of age, but the symptoms were present for a longer period of time (median 4 vs. 2 days, *p* = 0.015), although the duration of fever prior to hospitalization did not differ between the groups, neither did the length of stay. Thrombocytopaenia was observed in 11.5% of influenza B cases, 8.4% influenza type A, and 6.7% mixed influenza A and B. Thrombocytosis was seen mainly in influenza type A (9.2%), followed by A and B (6.7%), and type B (4.8%) infection (Table 2).

The median MPV reached 9.7 fL (IQR: 9.2–10.3) in the whole study group, while the median MPV/PLT was 0.0403 (IQR: 0.0295–0.0527), and the median PLT/LYM ratio equaled 98.88 (IQR: 63.9–166). 

There was a positive correlation between the patients’ age and the MPV/PLT (rho = 0.44) and the PLT/LYM (rho = 0.27), and a negative correlation with the PLT (rho = −0.46), while the MPV did not correlate with the age (Table 3 and Figure 2). An internal correlation was then calculated within the aforementioned age groups; in children under 12 months, the rho was significant for PLT, MPV, and PLT/LYM (rho = −0.21, −0.45, and −0.33, respectively), while there was no internal correlation among those aged 12–23 months, a weak correlation for PLT/LYM among 24–59 months olds (rho = 0.16), and weak yet significant correlations for MPV, MPV/PLT, and PLT/LYM among those aged 5 years and more (rho = 0.19, 0.18, and 0.22, respectively). Among possible confounding factors, we observed no clinically relevant differences with regards to gender (see Appendix A) or hydration status assessed by pH, bicarbonate, and base excess (see Appendix A).

In a multivariate regression model, the abnormal platelet count was correlated with increased odds of complications (OR = 1.67, 95%CI: 1.03–2.7, *p* = 0.036), but thrombocytopaenia nor thrombocytosis alone did not significantly increase the odds (Table 4). The abnormal platelet count in general (OR = 1.89, 95%CI: 1.16–3.07, *p* = 0.01), and thrombocytosis in specific (OR = 3.64, 95%CI: 1.87–7.08, *p* < 0.01) were significant with regards to LRTI. Thrombocytosis (but not abnormal platelet count) was related to an increased risk of radiologically/ultrasound-confirmed pneumonia (OR = 2.15, 95%CI: 1.042–4.44, *p* = 0.038). Platelet count abnormalities did not correlate with the presence of AOM. On the other hand, for the prediction of antibiotic use, only a lowered platelet count was significant (OR = 2.41, 95%CI: 1.08–5.4, *p* = 0.031), while the thrombocytosis and the abnormal platelet count remained insignificant. Similarly, children with thrombocytopaenia were at higher risk of a longer (than median) hospital stay (OR = 3.03, 95%CI: 1.36–6.74, *p* < 0.01). None of the platelet count abnormalities correlated with the tertiary care center transfer.

Furtherly, a regression model was run in the aforementioned age groups. Thrombocytosis was related to an increased odds ratio of pneumonia only in children under 1 year of age (OR = 3.79, 95%CI: 1.46–9.8, *p* < 0.01), but the odds were insignificant in the remaining age groups. Similarly, LRTI odds were increased in children under 1 year of age (OR = 4.22, 95%CI: 1.71–10.38, *p* < 0.01), but the odds in the other age groups remained insignificant. 

Children with complications were younger (median 28 vs. 38 months, *p* = 0.01), had a longer history of signs and/or symptoms (3 days vs. 2 days, *p* < 0.01), and fever (3 days vs. 1 day, *p* < 0.01), required longer LOS (6 vs. 5 days, *p* < 0.01), and presented with lower MPV (9.6 vs. 9.8, *p* = 0.019), a lower MPV/PLT ratio (0.038 vs. 0.041, *p* = 0.035), a lower PLT/LYM (87.93 vs. 110.07, *p* < 0.01), while the total platelet count did not differ between the groups (*p* = 0.075).

In the ROC analysis, the highest AUC calculated for the whole group was seen in the case of lowered PLT/LYM (AUC = 0.581, 95%CI: 0.531–0.632, *p* = 0.0016), followed by lowered MPV (AUC = 0.56, 95%CI: 0.51–0.61, *p* = 0.0177), and lowered MPV/PLT (AUC = 0.56, 95%CI: 0.5–0.61, *p* = 0.0382, Table 5). An age subgroup analysis revealed a significant correlation only in children under 12 months of age; in this age group, the highest AUC for the prediction of complications was observed for a lowered MPV/PLT ratio (AUC = 0.66, 95%CI: 0.56–0.75, *p* = 0.0017), followed by a lowered MPV (AUC = 0.65, 95%CI: 0.55–0.74, *p* = 0.0026), and PLT count (AUC = 0.61, 95%CI: 0.51–0.71, *p* = 0.0315). The cut-off values estimated with the Youden index exhibited the following sensitivity/specificity: 38 and 92% for MPV/PLT (at the cut-off of 0.02), 73% and 50% for MPV (at the cut-off of 9.9 fL), and 32 and 89% for PLT (at the cut-off of 456 thousand/µL, Table 6). All the analyzed parameters remained insignificant in the remaining age groups. 

Children with AOM had signs/symptoms of fever for a longer period of time (4 days vs 2 days, *p* = 0.029, and 3 days vs 2 days, *p* = 0.034), respectively, whereas the age did not differ. Among the platelet parameters, only the PLT/LYM ratio was significantly lower (PLT/LYM 87.62 vs. 101.85, *p* = 0.036), and adequately, a significant AUC in the ROC curve analysis was proved only for a lowered PLT/LYM (AUC = 0.57, 95%CI: 0.51–0.63, *p* = 0.025), while the other parameters remained insignificant. 

Children with LRTI were younger (23 months vs 37 months, *p* = 0.024), had a longer history of signs/symptoms and fever (4 days vs. 2 days, *p* < 0.01 and 3 days vs 1.5 days, *p* = 0.042, respectively), and required longer hospital treatment (7 days vs. 5 days, *p* < 0.01). They had a higher median platelet count (271 vs. 237, *p* < 0.01), lower MPV/PLT (0.036 vs 0.041, *p* < 0.01), and lower PLT/LYM (88.48 vs. 104.64, *p* = 0.015), while MPV did not differ between the groups.

The AUC for predicting LRTI equaled 0.59 (95%CI: 0.53–0.64, *p* = 0.029) for PLT count, and lowered MPV/PLT (95%CI: 0.53–0.65, *p* = 0.0017), and 0.57 (95%CI: 0.52–0.62, *p* = 0.01) for lowered PLT/LYM, while in the case of MPV, it remained insignificant. Similar to the prediction of complications, the age group analysis showed a significant AUC only in those under 12 months of age, and it equaled 0.7 for lowered MPV/PLT (95%CI: 0.6–0.8, *p* < 0.01), and 0.67 for the PLT count (95%CI: 0.57–0.77, *p* = 0.0013), while it was insignificant for the other two platelet parameters. The cut-off values showed a sensitivity and specificity of 44% and 93%, respectively; at the cut-off point of 0.02 for MPV/PLT, and 58% and 72% at PLT = 371 thousand/µL. 

Patients with a radiological/ultrasound confirmation of pneumonia presented signs/symptoms and fever longer (4 days vs 2 days, *p* < 0.01, and 3 days vs 1 day, *p* < 0.01, respectively), had longer LOS (7 days vs 5 days, *p* < 0.01), but showed only lower MPV/PLT (0.037 vs 0.041, *p* = 0.049), and a corresponding analysis did not reveal any statistical significance in terms of predicting pneumonia. According to the age groups, a significant relationship was seen for lowered MPV/PLT (AUC = 0.68, 95%CI: 0.56–0.8, *p* < 0.01) in children under 12 months of age (sensitivity and specificity of 48% and 88% at cut-off value 0.02, respectively), whereas none of the parameters were significant in the other age groups. 

The need for antibiotic therapy was related to a younger age (24 months vs 39 months, *p* < 0.01), longer history of signs/symptoms (3 days vs 2 days, *p* < 0.01), fever (3 days vs 2 days, *p* = 0.032), and a longer hospital stay (7 days vs 4 days, *p* < 0.01). Children with antibiotic treatment had a higher median platelet count (PLT 278 vs 229, *p* < 0.01), lower MPV (9.5 vs. 9.8, *p* < 0.01), and lower MPV/PLT (0.036 vs 0.043, *p* < 0.01), whereas the PLT/LYM was insignificant.

The PLT count (AUC = 0.62, 95%CI: 0.57–0.67, *p* < 0.01), a lowered MPV (AUC = 0.57, 95%CI: 0.52–0.63, *p* = 0.0058), and a lowered MPV/PLT (AUC = 0.62, 95%CI: 0.57–0.67, *p* < 0.01) showed a significant AUC in predicting the need for antibiotic treatment, while the PLT/LYM was insignificant. An age group analysis revealed a statistical significance for PLT, MPV, and MPV/PLT in those aged 1–2 years (AUC of 0.65, 0.68, and 0.66, respectively), and for PLT and MPV/PLT in children aged 2–5 years (AUC of 0.61 and 0.6, respectively). The PLT cut-off values of 284 thousand/uL and 231 thousand/uL in children aged 1–2 years and 2–5 years indicated a sensitivity/specificity of 64%/69% and 61%/58%, respectively. The MPV/PLT at the cut-offs of 0.03 and 0.04 showed 44%/81% of sensitivity/specificity and 51%/60%, respectively. In children aged 1–2, the MPV at the value of 9.5 fL had 67% and 63% of sensitivity and specificity, respectively. 

Children who required a transfer to tertiary care had presented signs/symptoms and had a fever for a longer period of time prior to hospitalization (6 days vs 2 days, *p* = 0.026, and 6 days vs. 2 days, *p* = 0.016), while there were no differences between their platelet parameters. The ROC curve analysis showed a significant AUC for the MPV (AUC = 0.77, 95%CI: 0.64–0.89, *p* < 0.01), while the other parameters were insignificant. At the cut-off value of 9.9 fL, the sensitivity was 100%, with a specificity of 56% (Figure 3).

## 4. Discussion

Our findings demonstrate that platelets play a significant role in influenza, and platelet parameters may add a significant value to the patient’s assessment, however, the results need to be treated with caution.

Firstly, platelet count abnormalities may suggest a more severe disease course, and the direction of the platelet count changes may vary depending on the pathology expected. While thrombocytosis corresponded with LRTI and pneumonia, thrombocytopaenia was more frequently observed in children who were given antibiotics and required a prolonged hospital stay. The relationship between the involvement of the lower respiratory tract (including, but not restricted to pneumonia) and thrombocytosis is in line with the higher platelet counts previously reported by Lee in 72 children with A/H1N1 pandemic influenza complicated by pneumonia, compared to those without pneumonia [16]. We observed higher odds of both pneumonia and LRTI in the whole study group, but the odds were driven mainly (and statistically significant only) by children under 1 year of age; in the youngest group, the odds ratios were higher than in the whole group (4.22 vs 3.64 for LRTI and 3.79 vs 2.15 for pneumonia, respectively). The influence of age specificities on the platelet number needs to be emphasized here and will be discussed in the further part of this section. 

Apart from the LRTI, another potential use of the platelet count may be a differentiation between bacterial and viral infections. While patients with antibiotics had a higher median platelet count, thrombocytopaenia, as a sole predefined platelet count abnormality, was related to increased odds of antibiotic use. A study on a group of adults hospitalized because of AH1N1 and CAP revealed higher platelet counts in those with a bacterial coinfection [15], but Hsing found in a cohort of 558 children hospitalized due to influenza that those with Streptococcus pneumoniae (one of the leading bacterial causes of pneumonia) suprainfection had lower platelet counts [32]. The presence of a bacterial coinfection may elicit a platelet count increase on one hand, on the other hand, influenza seems to decrease the platelet count substantially—the 2009 pandemic studies showed an important tendency towards a decreased PLT count in children with A/H1N1/pdm09 in Iran [33], or in both adult and minor patients in the US [34], and a thrombocytopaenia (<50,000) in some of the critically ill children in the UK [35]. However, this simplification might be hugely misleading (as seen in our series of patients) since the stage of pneumonia and its severity may be crucial here. Initial platelet sequestration in the lungs results in thrombocytopaenia in peripheral blood [36,37], and while platelet activation may even be beneficial, massive platelet aggregation might be lethal [37]. The pathophysiological pathways should be taken into account; a murine model of influenza type A showed a significant interaction between the neutrophils and platelets: while neutrophils are recruited towards the site of infection, platelets are being aggregated and a neutrophil extracellular traps (NET) are being produced. Thus, thrombin is activated, leading to local tissue damage [38]. To the contrary, an inhibition of the thrombin pathway or protease-activated receptor 4 (PAR4) protects from the damage [38]. An antagonistic effect on the platelet receptors (for example with the use of clopidogrel), combined with antiviral drugs in mice have been observed to reduce the neutrophil influx, NET release, and the improve survival rate [39]. 

A complex relationship between the platelet count and the clinical course is reflected in our study by the odds of complications; while an abnormal platelet count was correlated with an increased odds ratio, none of the abnormalities alone (i.e., thrombocytopaenia or thrombocytosis) were significant to predict the presence of complications. 

Another parameter analyzed, the MPV, acts differently with regards to the various endpoints; a decreased MPV, correlated with the presence of complications and antibiotic therapy, while an increased MPV was prognostic of the need for a tertiary reference center transfer. Of interest, the relationship between antibiotic use and the MPV was seen only in children aged 12–23 months, but it demonstrated the broadest AUC compared to other platelet parameters. Limited data regarding the MPV and respiratory complications are available; previous studies on pneumonia in adult patients showed that a high MPV [40], or its increase [41,42], correlates with mortality, although one of the studies revealed a reverse correlation between the MPV and pneumonia CURB-65 (Confusion, Urea, Respiratory rate, Blood pressure, age > 65 years) score [43]. The pediatric study by Karadag-Oncel analyzed 196 children with CAP (and 100 healthy counterparts) and indicated a lower MPV in children with pneumonia compared to the control group, yet, a higher MPV in the hospitalized cases rather than in outpatients [20]. Therefore, special interest should be put not even in the initial values of the MPV, but in the MPV changes during the disease; while a decreased MPV may be diagnostic of complications, a worse outcome may be predicted by its increase. 

Among the platelet parameters studied in this research, the MPV/PLT ratio seems to be the most versatile, in general, and to present the highest prognostic value in particular cases. Moreover, due to the inclusion of two parameters in the ratio (platelet count and their volume, PLT and MPV), the influence of age-related fluctuations on any of them is decreased. 

In our series of patients, the MPV/PLT ratio showed the ability to predict complications, and the need for antibacterial therapy, and had the most robust area under the curve for predicting complications (with LRTI as a specific condition), as well as antibiotic treatment. Our results showing a lowered MPV/PLT seem contradictory to the studies by Sayed and Golwala, who found a relationship between an increased MPV/PLT and higher mortality (in pediatric sepsis or general in-hospital mortality, respectively) [44,45], while no fatal outcome occurred in our study.

A similar diagnostic value of an increased MPV/PLT ratio was revealed in the studies by Han and Fei, who compared children diagnosed with infectious mononucleosis or influenza, respectively, against healthy controls [18,46]. However, our study setting was limited to children with laboratory-confirmed influenza who were hospitalized, and our analysis aimed to assess the risk of complications and the clinical course. Thus, taking into account the various confounding factors (activation of different inflammatory pathways, and possible bacterial suprainfections), our data does not need to be contradictory to the above studies, since it is restricted to children hospitalized due to influenza only. Moreover, it is worth stressing that attention should be also put on the direction of the MPV/PLT changes. 

Surprisingly, the MPV/PLT was the only platelet parameter that correlated in the ROC analysis with the presence of radiologically-confirmed pneumonia, and this was seen only in a subgroup of children under 12 months of age, whereas it remained insignificant in the other age groups and the whole study group. 

The need for antibiotic therapy correlated also with a decrease in MPV/PLT ratio, which is contradictory to the study by Lee, who reported higher MPV/PLT in patients with increased procalcitonin levels [47], that are known to be associated with increased likelihood of bacterial infection [48,49,50,51]. It needs to be underlined, however, that the endpoint in our protocol was based on antibiotic use, not on the microbiological or molecular confirmation of bacterial infection, nor in correlation with serum inflammatory markers. 

The fourth platelet parameter assessed in previous studies is based on a relationship between the PLT and lymphocytes. Fei and colleagues analyzed platelet parameters in terms of influenza type A (not B), including 191 children with laboratory-confirmed influenza A, compared their laboratory characteristics with children without influenza A who had or had not influenza-like illness (ILI) symptoms, and observed the highest diagnostic value (the largest area under the curve) for lowered LYM*PLT in distinguishing influenza A from non-influenza A ILI. It might probably be a disease-specific condition since influenza is related to the decrease in both lymphocytes and platelet count [18]. A relationship between the PLT and LYM was analyzed as a PLT to LYM ratio in a study by Kartal in children with pneumonia, and the PLT/LYM ratio was increased in children with CAP, especially in hospitalized cases [17]. To the contrary, we found a decreased PLT/LYM in children with complications, which included LRTI (but not radiologically-confirmed pneumonia), and AOM; in the latter case, it was the only significant platelet parameter.

There are several factors influencing platelet parameters. Kim et al. conducted huge research on platelet abnormalities during respiratory viral infections, which enrolled, inter alia, 4618 patients aged 0–9 years. An abnormal platelet count (thrombocytopaenia threshold <150,000/mm^3^ and thrombocytosis > 450,000/mm^3^) was seen in 24.3% of patients with influenza type B, and 11.4% with influenza A, while in our study, the frequency of an abnormal platelet count was similar: 17.6% in patients with influenza A, and 16.4% in children with type B, and, unexpectedly, in a lower percentage of children with mixed A and B coinfection (13.3%). A generally higher frequency of thrombocytopaenia rather than thrombocytosis reported in a study by Kim was not confirmed in our group of patients; we observed only a slight predominance of a lowered platelet count (9% vs. 8.2%) [19]. Nor did the much higher prevalence of thrombocytopaenia in those infected with influenza type B find confirmation in our study; the percentage was only slightly higher in type B versus A cases (11.5% vs. 8.4%), whereas Kim reported thrombocytopaenia in 20.9% of all influenza type B cases and in a much lower number of type A cases [19]. Correspondingly, the platelet count was higher in type A than B cases in a study by Oh [52]. 

The frequency of platelet abnormalities may depend on the study settings, host and virus characteristics, named not only by the influenza types but also by the subtypes/lineages. A study by Yan, for example, revealed that A/H3N2 patients had lower PLT counts than other influenza patients (a study conducted in patients aged 0.5–95 years) [53]. Our investigation did not focus on a direct comparison between influenza types, not the subtypes/lineages, but on the course and presence of complications in influenza, seeking practical tools for everyday practice. 

Interestingly, in the study by Kim, the highest prevalence of thrombocytosis was observed in infants (26.1%), then dropping rapidly, and children over 2 years of age presented an increasing tendency towards thrombocytopaenia [19]. Correspondingly, we found the vast majority of thrombocytosis cases in children under 24 months of age (85%, 34 out of 40), while thrombocytopaenia was observed mainly in older children (88.6% in those over 2 years old). 

In our study, we also analyzed the relationship between the patient’s age and the PLT number and found a moderate negative correlation (rho = −0.46). The recognition of age-related specificities is mirrored in the statistical significance of AUCs in the age groups. In the majority of cases, in children under 12 months of age this relationship was strong enough to result in statistical significance for the whole group, and, consequently, AUCs showed the highest values in this particular age group. An analysis of platelet parameters should be thus considered mostly in the youngest group of patients. 

The group of patients included in our research presented with complications in more than 42% of cases, which seems to be similar to other groups of hospitalized children with influenza. The frequency of radiologically-confirmed pneumonia in our study (over 23%) is in line with the 2009 pandemic analysis, which revealed pneumonia in over 20% of pediatric patients with A/H1N1pdm09 [12], and is almost 2-fold higher than the percentage observed by Peltola (10–12%) [4]. AOM, on the other hand, was reported by Peltola in 22–30% of children hospitalized due to influenza, while in our group it was slightly lower (18%); however, those discrepancies may result from different groups of patients, comorbidities, virus characteristics, and methodological variations in protocols [4]. The use of antibiotics (38%) in our series of patients is not low, but the percentages are generally similar to those reported, for example, during the 2009 pandemic (45%) in a huge group of patients (*n* = 6381 in the analysis by Muthuri) [12], reflecting the problem with the attempts to reduce the use of antibacterials; although a very restrictive policy on the use of antibacterials is implemented at our Department of Pediatrics, antibiotics remain necessary in many cases. 

There are several important strengths and limitations to this study. The research succeeded to gather a huge pediatric group of patients hospitalized due to laboratory-confirmed influenza, which, in our belief, may be considered representative. However, it is only a single-center study, and further multicenter studies are required. Secondly, its retrospective character did not allow the PLT and/or MPV to influence the clinical decision-making process, on the other hand, the clinical course was not influenced by the study, especially in terms of antibiotic therapy. Thirdly, we did not assess platelet distribution width (PDW), which might add more information, since it is changing especially under pathological conditions [54]. We did not perform an in-depth analysis of the underlying mechanisms of the platelet parameters’ alterations which were not included in the study protocol; nevertheless, this gap in knowledge needs to be filled. We also chose the use of antibiotics, not the confirmation of a bacterial infection as the endpoint, although the use of antibiotics (although restricted due to the local antibiotic policy) is not a synonym for an actual need for antibacterial therapy, it rather represents a practical clinical approach in which sensitivity of the methods used to diagnose bacterial infection is limited. The lack of a radiological confirmation of pneumonia decreased the number of participants by a low number of cases only, but the true frequency of pneumonia has not been assessed by an optimal protocol that would consist of an X-ray performance in each patient, yet such a practice would raise ethical concerns. A thorough clinical assessment was performed instead, decreasing the risk of missing pneumonia or LRTI cases. The study is limited to treating influenza as a homogenous condition; there are huge differences between influenza types, but also between subtypes and/or lineages, yet, due to practical implications and similarities in clinical presentation and management, we used this simplified approach. There are very important limitations related to platelet physiology, including the influence of the age and sex; in the pediatric group of patients the issue of platelet parameters’ changes with age is known, but also gender may influence the results—here, to diminish the influence of the patient-related factors, we performed the multivariate analysis which contained gender and used age subgroups to decrease the generalization bias [28]. Nonetheless, the age group division was made upon the influenza risk groups in order to facilitate future comparisons, but at the same time, the subgroups may influence the results’ interpretation as well. Similarly, the definition of thrombocytopaenia/thrombocytosis may be somehow controversial, nevertheless, it was made in line with previously published papers. Clinical differences between the patients need to be remembered, for instance, the status of hydration which plays a significant role in platelet parameters; here it was assessed by the capillary blood gas parameters and some minor differences were observed, which, in our opinion, had no further clinical implications. The time since the disease onset is another important factor in terms of the platelet pathophysiology, but in order to remain practical, we did not intend to classify patients into subgroups based upon the duration of the disease, but presented a practical approach with the inclusion of the whole cohort of the patients who were admitted to the hospital, which is a regular practice in pediatric studies. Moreover, a possible relationship with bacterial coinfections not only needs to be remembered but may also become a topic of future research. 

## 5. Conclusions

In conclusion, an abnormal platelet count was related to the increased odds of complications (including LRTI), thrombocytopaenia was related to the increased risk of longer LOS and antibiotic use, while thrombocytosis with the risk of LRTI and radiologically-confirmed pneumonia. The MPV/PLT ratio had the most robust AUC for predicting complications (with LRTI as a specific condition), and antibiotic treatment, while MPV correlated with tertiary reference center transfer. However, the calculated AUCs showed at the very best rather moderate usefulness of every single parameter, and a practical indication of cut-off values is complicated due to, inter alia, age-related differences. To conclude, platelet abnormalities in the course of influenza may contribute important data in the assessment of pediatric influenza patients, and further, prospective multi-center studies are necessary. A broad potential research value might be expected in the field of bacterial coinfections, and an in-depth analysis of the pathomechanism underlying platelet action is required.

## Figures and Tables

**Figure 1 diagnostics-13-00634-f001:**
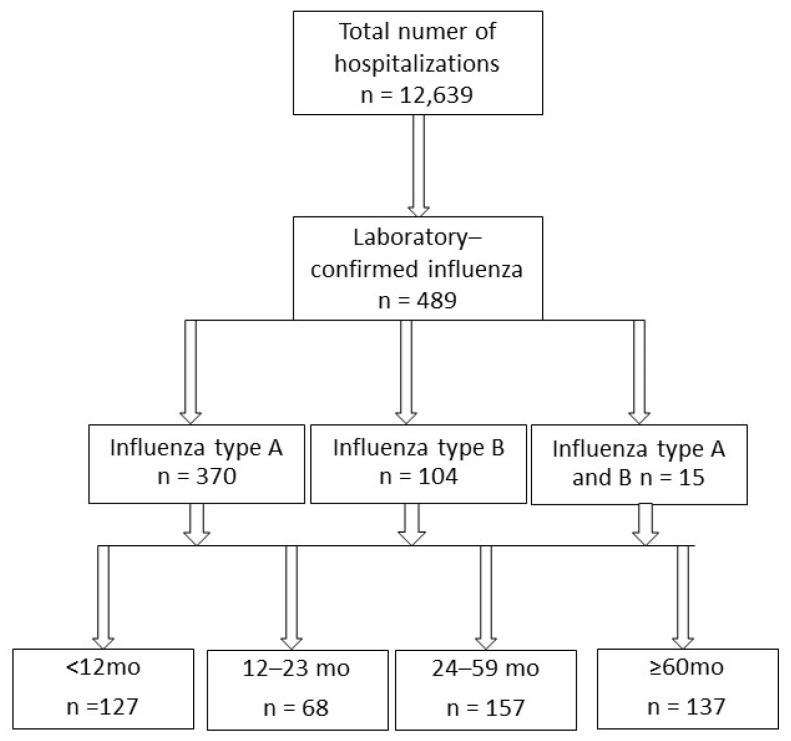
A flowchart of the patients in the study.

**Figure 2 diagnostics-13-00634-f002:**
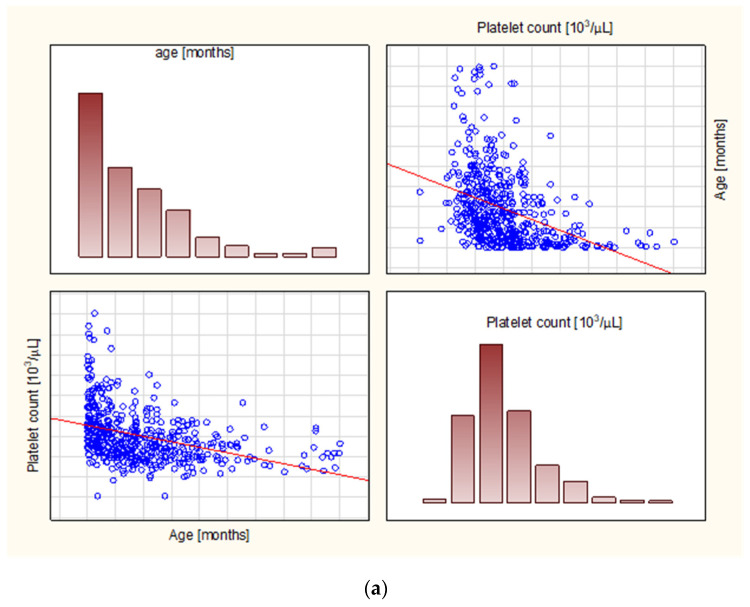
The correlation graphs between (**a**) age and platelet count, and (**b**) age and MPV/PLT ratio. In the diagram, (**a**) the age is showed on the x-axis, while the platelet count, (**a**) or the MPV/PLT ratio. The results are based on Spearmann’s rank correlation test.

**Figure 3 diagnostics-13-00634-f003:**
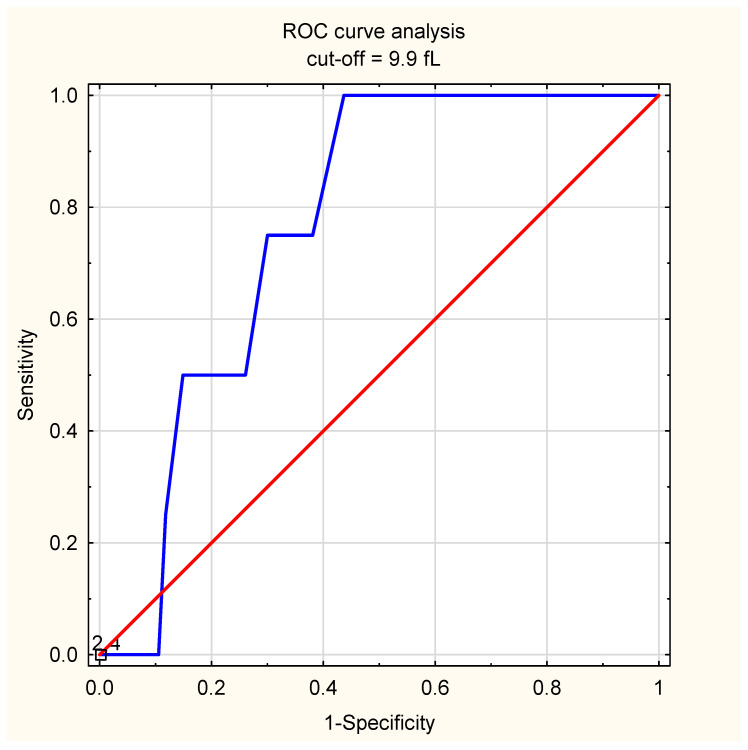
The results of the ROC curve analysis of MPV in the prediction of tertiary care unit transfer; the calculated area under the curve reached 0.77 (95%CI: 0.64–0.89, *p* < 0.01), and the optimal cut-off value was calculated with the Youden index; the blue line shows the ROC curve, while the red line is the reference line.

**Table 1 diagnostics-13-00634-t001:** Baseline characteristics of the study group (**a**) with the comparison between influenza types (**b**). Abbreviations: CRP—C-reactive protein, PCT—procalcitonin, WBC—white blood cells count, ANC—absolute neutrophil count, PLT—platelet count, MPV—mean platelet volume, MPV/PLT—mean platelet volume/platelet count ratio, PLT/LYM—platelet to lymphocyte ratio, *n*—number of patients, LQ—lower quartile, UQ—upper quartile. * The results of the comparison regard a comparison between the two most frequent influenza, type A and B; statistically significant results are bold.

(**a**)										
			** *n* **		**Median**		**LQ**		**UQ**	
duration of signs/symptoms prior to hospitalization [days]			489		2.0		1.0		5.0	
duration of fever prior to hospitalization [days]			489		2.0		1.0		4.0	
age [months]			489		34.0		11.0		63.0	
length of stay [days]			489		5.0		4.0		7.0	
CRP [mg/L]			488		5.42		1.39		15.94	
PCT [ng/mL]			456		0.18		0.11		0.36	
WBC [10 * 3/µL]			489		7.55		5.30		10.83	
ANC [10 * 3/µL]			489		3.40		2.03		5.83	
PLT [10 * 3/µL]			489		244.0		190.0		320.0	
MPV [fl]			487		9.7		9.2		10.3	
MPV/PLT			487		0.0403		0.0295		0.0527	
Hemoglobine [g/dL]			489		11.9		11.2		12.6	
Lymphocytes [10 * 3/µL]			489		2.40		1.46		4.02	
PLT/LYM			489		98.88		63.89		166.00	
(**b**)										
	**Influenza Type A (*n* = 370)**			**Influenza Type B (*n* = 104)**			**Influenza A and B (*n* = 15)**			
	**Median**	**LQ**	**UQ**	**Median**	**LQ**	**UQ**	**Median**	**LQ**	**UQ**	***p* ***
duration of signs/symptoms prior to hospitalization [days]	2.0	1.0	4.0	3.0	1.0	5.0	4.0	2.0	6.0	**0.032**
duration of fever prior to hospitalization [days]	2.0	1.0	4.0	3.0	1.0	5.0	2.0	1.0	6.0	**0.027**
age [months]	30.0	9.0	57.0	41.0	15.5	80.5	68.0	17.0	106.0	**0.001**
length of stay [days]	5.0	4.0	7.0	5.0	4.0	7.0	6.0	4.0	8.0	0.236
CRP [mg/L]	5.52	1.47	16.56	4.66	1.25	12.13	21.60	0.71	76.16	0.168
PCT [ng/mL]	0.19	0.11	0.36	0.13	0.08	0.30	0.35	0.17	0.52	**0.008**
WBC [10 * 3/µL]	7.76	5.48	10.83	7.34	4.79	10.08	11.50	3.20	16.80	0.169
ANC [10 * 3/µL]	3.36	2.03	5.75	3.53	2.07	5.76	4.82	0.68	12.33	0.790
PLT [10 * 3/µL]	247.5	191.0	329.0	234.5	183.0	291.5	254.0	196.0	306.0	0.059
MPV [fL]	9.7	9.2	10.3	9.8	9.1	10.4	10.0	9.4	10.1	0.533
MPV/PLT	0.0400	0.0284	0.0518	0.0421	0.0310	0.0562	0.0398	0.0319	0.0480	0.084
Hemoglobine [g/dL]	11.9	11.2	12.6	12.0	11.3	12.8	12.4	11.6	13.3	0.149
Lymphocytes [10 * 3/µL]	2.47	1.50	4.09	2.30	1.28	3.37	2.43	1.95	4.56	0.057
PLT/LYM	97.80	60.64	159.18	107.01	70.22	178.45	74.00	54.29	126.67	0.305

**Table 2 diagnostics-13-00634-t002:** A prevalence of platelet abnormal count with regards to the specific age groups and influenza types. Abbreviations: *n*—number of patients, yo—year(s) old.

Age Group	Thrombocytopaenia *n* (%)	Thrombocytosis *n* (%)
<1 yo (*n* = 127)	3 (2.4)	27 (21.3)
1–<2 yo (*n* = 68)	2 (2.9)	7 (10.3)
2–<5 yo (*n* = 157)	18 (11.5)	6 (3.8)
≥5 yo (*n* = 137)	21 (15.3)	0 (0)
Influenza type		
type A (*n* = 370)	31 (8.4)	34 (9.2)
type B (*n* = 104)	12 (11.5)	5 (4.8)
type A and B (*n* = 15)	1 (6.7)	1 (6.7)

**Table 3 diagnostics-13-00634-t003:** A correlation between platelet parameters (platelet count—PLT, mean platelet volume—MPV, mean platelet volume/platelet count ratio—MPV/PLT, platelet to lymphocyte ratio—PLT/LYM) and age with regards to the age groups; the results from Spearmann’s rank correlation test; results are showed as correlation coefficients; only statistically significant results are showed. Abbreviations: yo—years old, ns—statistically insignificant.

Age	PLT	MPV	MPV/PLT	PLT/LYM
Whole study group	−0.46	ns	0.44	0.27
<1 yo	−0.21	−0.45	ns	−0.33
≥1 yo–<2 yo	ns	ns	ns	ns
2–<5 yo	ns	ns	ns	0.16
≥5 yo	ns	0.19	0.18	0.22

**Table 4 diagnostics-13-00634-t004:** The relationship between platelet count abnormalities (including thrombocytopaenia and thrombocytosis) and the odds of complications and clinical course of influenza; the results of the multivariate regression model are expressed as odds ratios with 95% confidence intervals; only statistically significant results are shown. Abbreviations: LRTI—lower respiratory tract infection, OR—odds ratio, 95%CI—95% confidence interval, ns—statistically insignificant.

	Abnormal Platelet Count	Thrombocytopaenia	Thrombocytosis
Complications	OR = 1.67, 95%CI: 1.03–2.7, *p* = 0.036	ns	ns
Acute otitis media	ns	ns	ns
LRTI	OR = 1.89, 95%CI: 1.16–3.07, *p* = 0.01	ns	OR = 3.64, 95%CI: 1.87–7.08, *p* < 0.01
Pneumonia	ns	ns	OR = 2.15, 95%CI: 1.042–4.44, *p* = 0.038
Antibiotic therapy	ns	OR = 2.41, 95%CI: 1.08–5.4, *p* = 0.031	ns
Prolonged LOS	ns	OR = 3.03, 95%CI: 1.36–6.74, *p* < 0.01	ns
Tertiary care transfer	ns	ns	ns

**Table 5 diagnostics-13-00634-t005:** Usefulness of selected platelet parameters in prediction of influenza complications (**a**) and clinical course (**b**). The results of the ROC curve analysis are shown as the area under the curve with 95% confidence intervals; only statistically significant results are shown. The results are shown for the whole study group, and separately for age groups in which results were statistically significant. In the case of statistically significant AUCs under 0.5 (meaning a destimulating instead of stimulating effect of the verified parameter), the equation was reversed in order to present easily comparable results (i.e., AUC over 0.5), and an emphasis on the destimulating role was underlined by the word “lowered” preceding the result. Abbreviations: PLT—platelet count, MPV—mean platelet volume, MPV/PLT—mean platelet volume/platelet count ratio, PLT/LYM—platelet to lymphocyte ratio, yo—year(s) old, AUC—area under the curve, 95%CI—95% confidence interval, ns—statistically insignificant.

(**a**)				
	**PLT**	**MPV**	**MPV/PLT**	**PLT/LYM**
Complications	ns	Lowered AUC = 0.56, 95%CI: 0.51–0.61, *p* = 0.0177	Lowered AUC = 0.56, 95%CI: 0.5–0.61, *p* = 0.0382	Lowered AUC = 0.581, 95%CI: 0.531–0.632, *p* = 0.0016
Complications < 1 yo	AUC = 0.61, 95%CI: 0.51–0.71, *p* = 0.0315	Lowered AUC = 0.65, 95%CI: 0.55–0.74, *p* = 0.0026	Lowered AUC = 0.66, 95%CI: 0.56–0.75, *p* = 0.0017	ns
Acute otitis media	ns	ns	ns	Lowered AUC = 0.57, 95%CI: 0.51–0.63, *p* = 0.025
LRTI	AUC = 0.59 (95%CI: 0.53–0.64, *p* = 0.029	ns	Lowered AUC = 0.59 95%CI: 0.53–0.65, *p* = 0.0017	Lowered AUC = 0.57 (95%CI: 0.52–0.62, *p* = 0.01
LRTI < 1 yo	AUC = 0.67 (95%CI: 0.57–0.77, *p* = 0.0013	Lowered AUC = 0.7 95%CI: 0.6–0.8, *p* < 0.01	ns	ns
Pneumonia	ns	ns	ns	ns
Pneumonia < 1 yo	ns	Lowered AUC = 0.68, 95%CI: 0.56–0.8, *p* < 0.01	ns	ns
(**b**)				
	**PLT**	**MPV**	**MPV/PLT**	**PLT/LYM**
Antibiotic therapy	AUC = 0.62, 95%CI: 0.57–0.67, *p* < 0.01	Lowered AUC = 0.57, 95%CI: 0.52–0.63, *p* = 0.0058	Lowered AUC = 0.62, 95%CI: 0.57–0.67, *p* < 0.01	
Antibiotic therapy 1–2 yo	AUC = 0.65, 95%CI: 0.52–0.78, *p* = 0.0243	Lowered AUC = 0.68, 95%CI: 0.55–0.81, *p* = 0.0058	Lowered AUC = 0.66, 95%CI: 0.53–0.79, *p* = 0.0147	
Antibiotic therapy 2–5 yo	AUC = 0.61, 95%CI: 0.52–0.7, *p* = 0.0166		Lowered AUC = 0.6, 95%CI; 0.51–0.68, *p* = 0.0378	
Tertiary care center transfer		AUC = 0.77, 95%CI: 0.64–0.89, *p* < 0.01		

**Table 6 diagnostics-13-00634-t006:** Performance of platelet parameters (PLT—platelet count, MPV—mean platelet volume, MPV/PLT—mean platelet volume/platelet count ratio) in the predictions of influenza complications and tertiary care unit transfer. The results of the ROC curve analysis; the optimal cut-off values were calculated with the use of the Youden index. The results are shown as sensitivity, specificity, positive predictive value (PPV), and negative predictive value (NPV), with 95% confidence intervals (95%CI); only age groups with statistically significant values are shown.

<1 yo	Complications					
	PLT		MPV		MPV/PLT	
Optimal cut-off	456 × 10 * 3/µL		9.9 fL		0.02	
	value	95%CI	value	95%CI	value	95%CI
Sensitivity	32.08%	19.92–46.32%	73.08%	58.98–84.43%	37.74%	24.79–52.11%
Specificity	89.19%	79.80–95.22%	50.00%	38.14–61.86%	91.89%	83.18–96.97%
PPV	68.00%	49.78–82.00%	50.67%	43.67–57.64%	76.92%	58.97–88.55%
NPV	64.71%	59.98–69.16%	72.55%	61.52–81.37%	67.33%	62.31–71.98%
<1 yo	LRTI					
	PLT		MPV		MPV/PLT	
Optimal cut-off	371 × 10 * 3/µL				0.02	
	value	95%CI	value	95%CI	value	95%CI
Sensitivity	57.78%	42.15–72.34%			44.44%	29.64–60.00%
Specificity	71.95%	60.94–81.32%			92.68%	84.75–97.27%
PPV	53.06%	42.44–63.41%			76.92%	59.08–88.50%
NPV	75.64%	68.26–81.77%			75.25%	69.92–79.90%
<1 yo	Pneumonia					
	PLT		MPV		MPV/PLT	
Optimal cut-off					0.02	
	value	95%CI	value	95%CI	value	95%CI
Sensitivity					48.15%	28.67–68.05%
Specificity					87.88%	79.78%–93.58%
PPV					52.00%	35.91%–67.68%
NPV					86.14%	81.09%–90.00%
1–2 yo	Antibiotic					
	PLT		MPV		MPV/PLT	
Optimal cut-off	284 × 10 * 3/µL		9.5 fL		0.03	
	value	95%CI	value	95%CI	value	95%CI
Sensitivity	63.89%	46.22–79.18%	66.67%	49.03–81.44%	44.44%	27.94–61.90%
Specificity	68.75%	49.99–83.88%	62.50%	43.69–78.90%	81.25%	63.56–92.79%
PPV	69.70%	56.55–80.26%	66.67%	54.73–76.79%	72.73%	54.30–85.68%
NPV	62.86%	50.82–73.49%	62.50%	49.41–73.98%	56.52%	48.15–64.53%
2–5 yo	Antibiotic					
	PLT		MPV		MPV/PLT	
Optimal cut-off	231 × 10 * 3/µL				0.04	
	value	95%CI	value	95%CI	value	95%CI
Sensitivity	61.19%	48.50–72.86%			50.75%	38.24–63.18%
Specificity	57.78%	46.91–68.12%			60.00%	49.13–70.19%
PPV	51.90%	44.23–59.48%			48.57%	40.06–57.17%
NPV	66.67%	58.53–73.92%			62.07%	54.90–68.75%
0–18 yo	Tertiary Care Unit Transfer					
	PLT		MPV		MPV/PLT	
Optimal cut-off			9.9 fL			
	value	95%CI	value	95%CI	value	95%CI
Sensitivity			100.00%	39.76–100.00%		
Specificity			56.31%	51.76–60.79%		
PPV			1.86%	1.68–2.05%		
NPV			100.00%	undefined		

## Data Availability

Data are available on request from the authors.

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
