# Peer review of "Platelet Abnormalities in Children with Laboratory-Confirmed Influenza"

_diagnostics, 2023, doi:10.3390/diagnostics13040634_

Round 1

Reviewer 1 Report

The authors aimed to analyze prognostic value of platelet parameters (PLT, MPV, MPV/PLT, PLT/LYM) in children hospitalized due to influenza with regards to a prediction of complications (AOM, pneumonia and lower respiratory tract infection-LRTI), and clinical course of influenza (need for antibiotic treatment, need for tertiary care transfer, death).The study is novel and interesting.

- Introduction is clear.

- Methods are very well detailed and well structured

- Statistics is robust and adequate

- Results are relevant.

- Table and figures are very detailed and informative.

- Discussion is full of adequate critical insight.

Overall considered it is an excellent article and the authors should be congratulated.

Author Response

Dear Reviewer,

Thank You very much for Your favourable review and positive feedback, it increases the willingness to keep on working in the field of science!

Best regards,

August Wrotek

Reviewer 2 Report

Dr Wrotek and colleagues present an interesting draft focused on the estimation of platelets' basic parameters in children suffering from IAV/IBV infections. Since respiratory tract infections are emerging fields of medicine, all the novel findings are welcome to the field. 

I went very carefully through the whole paper and I would like to point out the things that are the major issues I found during this process. Please follow: 

- firstly, the PLT count during the first 5 years of life is constantly decreasing from ~350k to ~260k. It is a well-known phenomenon since ~1970. Table 2 just confirms these findings and does not add anything to the current state-of-the-art. Moreover, there are no guidelines given for the definition of thrombo-cytopenia/cytosis. Both of PLT abnormalities are marginal (<12%) and these results do not support the conclusions. 

- the correlation between PLT and age is known (as referred to above). Taking into account heterogenous PLT counts between IAV/IBV/IA+BV, no causative conclusions could be withdrawn. 

- the basic PLT parameters are their count, MPV, and PDW. The last one is missing in the paper. It is very unfortunate since PDW is so tightly connected with the age of patients. 

- I also wonder how the age-based division has been done. Can the Authors refer to any keystone paper showing similar patterns? Why do not set 0-7mo, 8-19 mo, 20-38 mo, and 39-78 mo intervals? What if the inclusion of 5+yo patients introduced lots of bias? Please keep in mind that the development of infants is strongly dynamic compared to 5.5 yo child. 

 - I miss the comparison/analysis of the studied parameters in terms of division based on sexes. Please be advised that female children are characterized by a higher number of circulating PLTs compared to males children. 

- What was the percentage of IA/BVs confirmation ONLY by RIDT? What is the reliability of used RILTs?

- What was the purpose of the strong focus on antibiotic therapy? I guess it was secondary therapy for bacterial co-infections. What was the standard antiviral therapy used? 

- How was the patient's hydration monitored? For young patients it is crucial for correct evaluation of any blood cells count, 

- I am also afraid that the results are highly biased due to the fact that IAV patients showed up in the clinic after 2 days (median) of symptoms and for A+B infection after 4 days. This is not a fair analysis. 

- Also values of age [months] for these three subgroups have P<0.001 which means that the conclusions might be highly biased.

- CRP values are extremely heterogeneous. There is a lack of information on how it was assayed. The same for PCT.

- Figure 2 has mislabeled/nonlabeled Y/X axis making them non-readable.

- Please avoid mixing American English with British English. I strongly suggest native speaker polishing. 

Author Response

Dear Reviewer!

I would like to express my personal gratitude for Your scrupulous review (and for Your patience)! Although it is not favourable at the moment, I hope to be able to explain all the doubts. Below, I would like to address all the suggestions that You have mentioned one-by-one:

- firstly, the PLT count during the first 5 years of life is constantly decreasing from ~350k to ~260k. It is a well-known phenomenon since ~1970. Table 2 just confirms these findings and does not add anything to the current state-of-the-art. Moreover, there are no guidelines given for the definition of thrombo-cytopenia/cytosis. Both of PLT abnormalities are marginal (<12%) and these results do not support the conclusions.

The question of age presents some problems in interpretation of the platelet results. Firstly, I totally agree, the issue of changes in the number of platelets in children is well-known, so are the tendencies in the number of platelets (rather than the exact range norms, but I will discuss it later). Table 2 confirms this well-known fact (the frequency of thrombocytosis decreases, while thrombocytopaenia increases with age), and we do not claim do discover new tendencies in platelet parameters, it is not a study design to verify such a question, we just report the frequencies of platelet count abnormalities in the study subgroups. Nevertheless, we find this table necessary, but only as a general information about the study group! The most controversial decision we made might be the sequence of the results’ presentation- we presented the multivariate regression model results at the beginning. The major reason is the fact that the model included age and gender as variables, attenuating thus the influence of both of them.

Regarding the definition of thrombocytopaenia/thrombocytosis- I personally do not agree with those age limits, however, we need to refer to the existing studies in the area! In the MM section we referred to the study that included the highest number of patients (Kim et al.) and that study used those definitions. Another (huge) study on the topic included a mixed group of patients (less than 45% of children, the rest were adults) used also 150,000 to define thrombocytopaenia, and 350,000 for thrombocytosis (compared to what is known in paediatrics, the range seems too low). We did not classify the severity of platelet count abnormalities as it would generate more divisions and subgroups. We did not discuss the range values due to the length of the manuscript. To conclude this paragraph, we just used definitions from the most significant study in the field.

Most of all, we do not claim that thrombocytopaenia/thrombocytosis (regardless of their definitions) causes or equals pneumonia/complications/etc. We just showed that in our group of patients, the odds are increased (odds ratio is more adequate here than risk ratio, for example), and we also show the results of the ROC analysis due to different methodology, and the conclusions can be drawn based upon those two types of complementary studies. Another reason to present OR first is simplicity of interpretation of the results. We conclude that “platelet abnormalities in the course of influenza may contribute important data in the assessment of paediatric influenza “. We believe the statement is true, based on the results, and puts attention to the assessment of platelet counts which seems to be a forgotten laboratory analysis, taken into account very low number of studies that focused platelets in paediatric influenza.

- the correlation between PLT and age is known (as referred to above). Taking into account heterogenous PLT counts between IAV/IBV/IA+BV, no causative conclusions could be withdrawn.

That would be true if we considered influenza A to be different from B or A and B coinfection. However, facing the choice of classifying influenza as a homogenous infection (which is a huge simplification, please see below) or distinguish influenza subtypes/lineages at least (which is completely impractical at the moment), the simplified approach is widely used, and this is due to similarities in clinical presentation and management (risk groups and treatment). Alike, we made the analysis in the whole group and age subgroups, but the division was not based on influenza types.

- the basic PLT parameters are their count, MPV, and PDW. The last one is missing in the paper. It is very unfortunate since PDW is so tightly connected with the age of patients.

We had a long discussion on this topic at the Ethics Committee. PDW in fact might be very useful, especially under pathological conditions when it does not follow changes in MPV, and might add important information. Nevertheless, this is a retrospective study which aimed to verify previously described platelet parameters in a huge group of patients. Unfortunately, there is no literature available (to our knowledge) on PDW changes in children with influenza, and only parameters mentioned in the introduction section had been studied. The Ethics Committee disadvises studying completely “new” parameters (i.e. without any proofs in the literature) in a retrospective manner. On the other hand, without reliable retrospective studies, there will not be prospective studies. Lack of PDW is a flaw, yet it was agreed at the Ethics Committee to verify only those parameters. 

We added an information in the limitations section, since, as You mentioned, PDW might be an important parameter.

- I also wonder how the age-based division has been done. Can the Authors refer to any keystone paper showing similar patterns? Why do not set 0-7mo, 8-19 mo, 20-38 mo, and 39-78 mo intervals? What if the inclusion of 5+yo patients introduced lots of bias? Please keep in mind that the development of infants is strongly dynamic compared to 5.5 yo child.

I totally agree, the age influence may seriously bias the results. For this reason we decided to create age groups.

There are 3 major age groups regarding the risk of severe influenza course and complications: under 2yo, 2-5yo, and ≥5yo (https://publications.aap.org/pediatrics/article/148/4/e2021053744/183303/Recommendations-for-Prevention-and-Control-of?autologincheck=redirected ). This approach is widely used in risk assessment and was the basis for the creation of the age groups. Additionally, since the stud by Kim (https://pubmed.ncbi.nlm.nih.gov/25545354/) described the highest frequency of platelet count abnormalities (in the course of influenza) in the first year of life, with important differences between the first and the second year of life, we finally created 4 age groups (with a separate <1 and 1-2 yo groups). I added an adequate description in the MM section.

I would like to underline that the analysis was first performed in the whole study group, and then in the subgroups.

The validity of the proposed age groups creation is confirmed by the number of patients in the age groups (statistically taken, the groups are well comparable), and by the reduction of correlation coefficients for all of the parameters except for MPV; moreover, in the case of PLT/LYM we managed to show different tendencies in various age groups, thing that would not be seen in the whole group analysis. Additionally, MPV showed significant fluctuations in the <1yo group. Nevertheless, if there were no age groups, the fluctuations in MPV would be hidden. I also added a short comment in the methodology section, emphasizing continuous changes of platelet parameters.

Mathematically taken, a step-by-step regression or other indirect methods (https://pubmed.ncbi.nlm.nih.gov/23412879/)  could indicate groups more accurately, yet the number of patients is too low for this purpose.

 - I miss the comparison/analysis of the studied parameters in terms of division based on sexes. Please be advised that female children are characterized by a higher number of circulating PLTs compared to males children.

It is a well-known phenomenon outside the neonatal age. Gender was included into multivariate logistic regression model (which was presented in the results first), however in the ROC analysis we did not assess children separately in gender-based subgroups. In order to clarify all the doubts, I put in the supplement section an additional analysis on gender-based differences with regards to the platelet parameters, and it turns out that only MPV differed between the sexes in the whole group (the p value=0.042 is close to statistical significance, yet it is significant in the whole study group). When divided into age-subgroups, we observed differences in terms of MPV for those aged 1-2 yo (p=0.0459), PLT count and MPV/PLT in those>5yo (p=0.026 and p=0.0489, respectively), which seems to have no practical meaning (I added such a statement).

The multivariate model contains gender as variable.

- What was the percentage of IA/BVs confirmation ONLY by RIDT? What is the reliability of used RILTs?

Only patients with laboratory-confirmed influenza were included, so we did not address in the MM section the topic of sensitivity/specificity of the tests. The specificity of RIDTs is known to be very high (it reaches 98%-99%), as can be found here, for example:

https://pubmed.ncbi.nlm.nih.gov/20156902/,

https://pubmed.ncbi.nlm.nih.gov/27139081/,

https://pubmed.ncbi.nlm.nih.gov/26011531/

The true problem, especially in the paediatric group of patients, is a low sensitivity of RIDTs, so there might have been some missed cases (due to false negative RIDT results), but those patients were not included. Certainly false positive/negative results of RT-PCRs may occur as well. We did not put information on various diagnostic methods to make the paper reasonably short (or rather not too long). For curiosity, I checked in the database and, depending on the year, 60-70% of the cases had RT-PCR confirmation, but as stated before, we do not perform both RIDT and PCR in every patient, neither did we compare platelet parameters against diagnostic method that was used (even if there was a statistical significance, no causative link should be expected). Moreover, only patients with clinical suspicion of influenza are tested towards influenza; thus, we consider a positive RIDT as true positive, while negative might be false negative and in suspected cases the PCR should be performed. This is in line with general recommendations by the CDC.

- What was the purpose of the strong focus on antibiotic therapy? I guess it was secondary therapy for bacterial co-infections. What was the standard antiviral therapy used?

Another very interesting and possibly significant point- according to the guidelines (AAP, for example, https://publications.aap.org/pediatrics/article/148/4/e2021053744/183303/Recommendations-for-Prevention-and-Control-of?autologincheck=redirected, or CDC, https://www.cdc.gov/flu/professionals/antivirals/summary-clinicians.htm) any patient hospitalized due to influenza should be given an antiviral treatment. And all the patients hospitalized due to influenza at our ward obtained antiviral treatment (oral oseltamivir in adequate dosage dependent both on age and weight). Some of the patients had their treatment started prior to hospitalization (although the percentage is marginal), nevertheless, since oseltamivir has not been related to significant changes in platelets parameters, we did not put it as variable in the analysis.

On the other hand, there might be differences in platelet parameters between the patients with and without bacterial coinfection and this could be an important diagnostic trace for the future studies. For this reason, assuming that bacterial coinfections may influence platelet parameters, we chose to verify those with vs. without antibacterial treatment.

Taken into account a low sensitivity of diagnostic methods in confirmation of bacterial coinfection (sensitivity of blood cultures in pneumonia, for example, is ca. 0.2-5%, molecular methods increase sensitivity 2.5-3.5-fold, but  still it remains low, even pleural fluid cultures are negative in around 50% of the cases) we chose this reverse approach based on an assumption that if a patient did not obtain antibiotic then no (significant) bacterial infection was present. This approach was used for the first time in the area  of paediatric pneumonia, since there are many problems with aetiology confirmation in pneumonia.

In the case of influenza, pneumonia and otitis media are the most frequent bacterial coinfections, but, again, it is hard to prove bacterial origin of the infection. Of course, an inflammatory marker-based approach could be used, but CRP nor PCT do not reflect bacterial infection with enough specificity or PPV, its strength is rather based on high NPV.

We added the information on the antiviral treatment topic in the MM section, but we did not add the discussion on the topic due to paper length.

- How was the patient's hydration monitored? For young patients it is crucial for correct evaluation of any blood cells count,

Another significant point, thank You! Although none of the previous studies verified the level of dehydration, and with the assumption that the study group contains a high number of patients, this could be skipped, nevertheless, we want to discuss the topic in-depth and performed additional analyses.

To make the issue more complicated, the vast majority of the clinical scales/scores for dehydration is physician-dependent, and they are mainly 3-level scales, thus, none of the clinical scales/scores was used here. To verify a possible influence of dehydration we chose CBG (capillary blood gas) results as the most objective and reliable study (we added reference). The CBG reflects the arterial blood values very strictly (with exception for pO2 and with exception for patients with hypotension- luckily none of our patients had hypotension). We used 2 methods of comparison- first, we checked for correlations between CBG parameters and platelet parameters, showing that although BE in the whole group correlated weakly (rho ca. 0.15-0.17) for PLT count, MPV and MPV/PLT, the subgroups were well created and the correlations in the subgroups were observed only in the case of MPV <1yo (rho=0.24 for BE), none correlations in the 1-2 yo group, more significant correlations in 2-5 yo (for PLT count and MPV/PLT rho=-0.27 and 0.25, respectively for both HCO3 and BE),and none significant correlations in >5yo.  In the second analysis, the patients (in the whole group and in the subgroups) were divided based upon presence/absence of abnormal platelet count/thrombocytosis/thrombocytopaenia, and the results here are less significant- they are showed  in the supplementary materials. 

- I am also afraid that the results are highly biased due to the fact that IAV patients showed up in the clinic after 2 days (median) of symptoms and for A+B infection after 4 days. This is not a fair analysis.

It would be unfair, indeed, however this is only median value and the group was included as whole- we did not perform analysis separately for influenza types (as mentioned above). Certainly, there are differences between influenza type A and B or coinfection, but (and I dare to say we have huge experience in this field) there are no clinically relevant methods (signs, symptoms, anything) that could distinguish one from another (except for testing). We showed the information on influenza types only for transparency, but they have no further meaning and A vs B approach should not be present anymore. There are more significant differences between A/H3N2 and A/H1N1 than between A and B in general; not only types, but subtypes (and genetics!!!) or lineages do matter! Regarding the time of signs/symptoms- the p value is relatively high (close to the statistical significance), and the p value (as stated in the manuscript) refers to A vs B comparison only, IQRs are overlapping; more significant differences would be in a comparison against AandB coinfection, but this have no practical significance. The results are shown in the Table 1b which shows baseline characteristics and is expected to facilitate any future comparisons, if any researcher would be interested in influenza types differences. Table 2, on the other hand, again for transparency reasons only, shows the frequency of thrombocytopaenia/thrombocytosis with regards to influenza types, but we did not calculate ORs for influenza types separately. I want to emphasize that a comparison between A-B-AB types was not the aim of the study. Nevertheless, we felt obliged by the study by Kim to discuss the observed differences in platelet count abnormalities with regards to influenza types.

We believe (as clinicians) that the science needs to be easily translatable into practice. Since the majority of the physicians (even in hospital settings) do not have access to diagnostic tools that would distinguish subtypes/lineages (H3N2 from H1N1 or Victoria from Yamagata lineage, for example), and only around 25% of the subtypes have been defined in our group of patients, we did not put subtypes/lineage analysis neither. There is a short comment on this topic in the discussion section, if You consider it not enough, we will be more than happy to add a paragraph to the discussion. The time since disease onset, on the other hand, is important for platelet parameters, but in order to remain practical we did not intend to classify patients into subgroups based upon duration of the disease- from practical point of view, no one would literally wait with results’ assessment until, let’s say, 3rd day of the disease just because the studies standardized platelet parameters for this particular duration of symptoms. A patient that enters a hospital ward, is the patient that needs to be assessed at this very moment and this approach is a regular practice in paediatric studies.

- Also values of age [months] for these three subgroups have P<0.001 which means that the conclusions might be highly biased.

As explained above, influenza types did not influence the analysis of the platelet count abnormalities, so type-related differences in age did not influence the results. We would be delighted to be able to show analysis on influenza subtypes (rather than types, which is currently not accepted) influence on platelet counts, but this was not the aim of the study.

- CRP values are extremely heterogeneous. There is a lack of information on how it was assayed. The same for PCT.

Of course, pleases excuse me the lack of this information in the MM section, I completed the methods, including limits of detection. We did not define a potential bacterial coinfection by inflammatory markers (CRP or PCT above a cut-off level as a substitute for bacterial infection), we did not define the bacterial coinfection as positive blood cultures/molecular study result due to limitations of those studies (although we conduct studies both on CRP and PCT in influenza). We chose (as explained above) the antibiotic-based model due to practical application.  

- Figure 2 has mislabeled/nonlabeled Y/X axis making them non-readable.

Please excuse, in fact the graph was made with Statistica and I did not notice the lack of labelling. Now it is corrected.

- Please avoid mixing American English with British English. I strongly suggest native speaker polishing.

 I did my best, now it is corrected by a translator (not native speaker, but academic teacher).

Thank You for so in-depth, constructive review, recently it does not happen too often, to tell You the truth. I believe with Your help we were able to improved our manuscript to make it acceptable.

Thank You!

Reviewer 3 Report

This study aimed to analyse prognostic value of platelet parameters in children hospitalized due to laboratory-confirmed influenza. 

- You mentioned in the introduction with references the following statements: "An increased platelet to lymphocyte ratio (PLT/LYM) has been suggested to have a diagnostic value in distinguishing community-acquired pneumonia (CAP) from healthy controls [17]. Moreover, it showed a potential prognostic value since inpatients had higher PLT/LYM compared to outpatients [17]. Platelet count may possibly be useful in differentiating between influenza and bacterial coinfection; in a series of patients hospitalized due to AH1N1 and CAP, higher PLT were observed in those with bacterial coinfection [15]". So in what way your study is novel? When most of the prognostic factors are already known, why di you conduct this study?

- Correct the spelling of "thrombocytopaeni" in the key words

- Line 46 - early implementation of the treatment (optimally within 48 hours since the first symptoms) is crucial for shortening duration of symptoms. Yours is a retrospective study. So how could you relate this statement with your study?

- You categorized children into: children under 12 months of age, 12-23 months of age, 24-59 months old, and 60 or more months. Until how many age, did you consider as children? Until which age, did you include in your study? Explain that in the methodology.

- Was there any difference in the prognostic factors based on gender?

- Line 449 - Taken into account  various confounding factors (activation of different inflammatory pathways, possible bacterial suprainfections), our data does not need to be contradictory to the above studies, since it is restricted to children hospitalized due to influenza only, although an emphasis should be put on the direction of MPV/PLT changes.

Explain this statement and the need for this statement.

Author Response

Dear Reviewer!

Thank You very much for Your review, the comments and the corrections You suggested! I believe that after improvements the manuscript presents much more value!

We would like to address these issues You raised one by one:

- You mentioned in the introduction with references the following statements: "An increased platelet to lymphocyte ratio (PLT/LYM) has been suggested to have a diagnostic value in distinguishing community-acquired pneumonia (CAP) from healthy controls [17]. Moreover, it showed a potential prognostic value since inpatients had higher PLT/LYM compared to outpatients [17]. Platelet count may possibly be useful in differentiating between influenza and bacterial coinfection; in a series of patients hospitalized due to AH1N1 and CAP, higher PLT were observed in those with bacterial coinfection [15]". So in what way your study is novel? When most of the prognostic factors are already known, why di you conduct this study?

Thank You very much for putting attention to it, it seems we did not present our rationale good enough, neither did we focus on the meaning of the study. While, in fact, it is not a novelty per se, yet in many aspects it presents a novel approach (otitis media as an end-point or treating LRTI as a whole, not only focusing on pneumonia, antibiotic treatment). As You mentioned all of the parameters have been assessed in few studies, the problem is that the data on the topic is very scarce and studies performed on the groups with high-number of patients are needed. When one takes a deeper look into the studies cited above, it turns out that the studies have been performed in different settings (or were not even paediatric studies), had different end-points or were performed in rather lower number of patients (varying between 72 and 164, while we included 489 laboratory-confirmed cases).

- Correct the spelling of "thrombocytopaeni" in the key words

Of course, corrected.

- Line 46 - early implementation of the treatment (optimally within 48 hours since the first symptoms) is crucial for shortening duration of symptoms. Yours is a retrospective study. So how could you relate this statement with your study?

Although there is no direct relationship with the aim of the study, we found this information relevant here to give the reader a general knowledge on the topic of influenza, and treatment possibilities (and risks related to the failure of the treatment). According to the CDC guidelines (and, for example, the Polish ones too) children who are hospitalized due to influenza or are at higher risk of complications require antiviral treatment (in fact regardless of the time that have passed since the symptoms onset, but optimally as soon as possible). Moreover, after suggestion from another reviewer, we added the information on the treatment of the patients (in fact all the patients received oral oseltamivir in adequate doses)- for this moment, oseltamivir has not been related to significant platelet changes, nevertheless, for future studies, this information should be added.

If You find the sentence needless, it can be certainly cut out, but I think that for a reader without in-depth knowledge in the field, it is valuable information.

- You categorized children into: children under 12 months of age, 12-23 months of age, 24-59 months old, and 60 or more months. Until how many age, did you consider as children? Until which age, did you include in your study? Explain that in the methodology.

Of course, we added the information in the material/method section- we considered children to be patients under the age of 18 years; in Poland we consider a patient to be paediatric patient right under this age, but the approach varies by+/-1 year. Thank You for noticing it!

The age-relationship seems to be crucial both for platelet parameters and influenza course.

For this reason we decided to create age groups. Since there are 3 major age groups regarding the risk of severe influenza course and complications: under 2yo, 2-5yo, and ≥5yo (https://publications.aap.org/pediatrics/article/148/4/e2021053744/183303/Recommendations-for-Prevention-and-Control-of?autologincheck=redirected ). This approach is widely used in risk assessment and was the basis for the creation of the age groups. Additionally, since the stud by Kim (https://pubmed.ncbi.nlm.nih.gov/25545354/) described the highest frequency of platelet count abnormalities (in the course of influenza) in the first year of life, with important differences between the first and the second year of life, we finally created 4 age groups (with a separate <1 and 1-2 yo groups). I added an adequate description in the MM section.

- Was there any difference in the prognostic factors based on gender?

Very significant remark! Gender may influence platelet parameters, thus it was included into multivariate logistic regression model (which was presented in the results first), however in the ROC analysis we did not assess children separately in gender-based subgroups. In order to clarify all the doubts, I put in the supplement section an additional analysis on gender-based differences with regards to the platelet parameters, and it turns out that only MPV differed between the sexes in the whole group (the p value=0.042 is close to statistical significance, yet it is significant in the whole study group). When divided into age-subgroups, we observed differences in terms of MPV for those aged 1-2 yo (p=0.0459), PLT count and MPV/PLT in those>5yo (p=0.026 and p=0.0489, respectively), which seems to have no practical meaning (I added such a statement). We added this analysis in the supplementary materials. Nevertheless, the multivariate model contains gender as variable.

- Line 449 – Taken into account  various confounding factors (activation of different inflammatory pathways, possible bacterial suprainfections), our data does not need to be contradictory to the above studies, since it is restricted to children hospitalized due to influenza only, although an emphasis should be put on the direction of MPV/PLT changes.

Explain this statement and the need for this statement.

The statement discusses the differences in the direction of MPV/PLT changes- while we observed statistically significant value for lowered MPV/PLT ratio,  some of the authors observed that increase in MPV/PLT has a prognostic/diagnostic value. The statement is a generalisation, since the other studies referred to mortality or diagnostic value (when diagnosing influenza in comparison with healthy controls), and we wanted to emphasize the differences between the study settings that might explain the observed differences in the results. I rephrased the sentence, splitting it into two separate (clearer) sentences.

Best regards,

August Wrotek

Round 2

Reviewer 2 Report

Kudos to the Author for a truly well-done handling of the critique from the first round of the peer review process. 

The Author in a very good manner addressed my majors as well as performed additional analysis as requested. 

This discussion shows that the Author is a well-focused clinician with lots of good insights into pediatric science. However, the novelty is not sky-high, these clinical observations might be useful for other clinicians in the future, thus, I am leaning towards the acceptance of the paper.  But, I still have some minors: 

- please make sure that new supplementary files are correctly cited within the manuscript. 

- please put some of my major (up to the Author which ones) comments as the limitations in the discussion section. 

- Please also make sure that the supplementary files will be available for the Readers since there are much more relevant data. 

Thanks for the great scientific talk and wish you all the best. 

Author Response

Dear Reviewer,

Thank You for all Your comments, which hugely improved our paper! I realize that our manuscript is not very novel, although (as clinicians) we found it very important, since the topic of the platelets’ function and simple blood analysis results’ interpretation seems to be underestimated; without such a studies, I am afraid, the topic will remain forgotten for a longer period of time, until, hopefully, an animal model gives more insights into the platelets’ pathology. Regarding Your comments:

  • I double-checked the citation of the supplementary materials- they are correct,
  • Certainly, I extended the limitations section and put all of Your major comments there (except for the RIDTs and diagnostic methods used for laboratory confirmation in order not to make the discussion excessively long)- they really help not only to understand but to interpret the results;
  • I will make sure that the supplementary materials are available (although I never had problems with MDPI as publisher, they always make available all the materials that are sent; just in case I added the titles of the tables at the end of the paper).

Thank You very much Your help and commitment!

Best regards,

August Wrotek

Reviewer 3 Report

The authors have satisfactorily answered all my queries.

Author Response

Dear Reviewer,

We are glad we were able to explain all the doubts, thanks to Your comments we had the possibility to improve our manuscript ! Thank You very much for Your favorable opinion, it means a lot to us!

Best regards,

August Wrotek